# Smart Material Handling Solutions for City Logistics Systems

Snežana Tadić , Mladen Krstić , Svetlana Dabić-Miletić and Mladen Božić *

Faculty of Transport and Traffic Engineering, University of Belgrade, 11000 Beograd, Serbia;
s.tadic@sf.bg.ac.rs (S.T.); m.krstic@sf.bg.ac.rs (M.K.)
* Correspondence: mladen.bozic@sf.bg.ac.rs

**Abstract:** Globalization, the growth of the world population, urbanization and the growth of the volume of the flow of goods have generated numerous problems in city logistics (CL). The opportunity to solve them is found in various fields by defining and implementing initiatives, concepts, measures, modern technologies and scenarios. The efficiency of the solution largely depends on the efficiency of logistics centers, which is one of the key subsystems of CL. The requirements for the reliable delivery of goods to customers in urban areas are conditioned by the efficiency their order fulfillment in logistics centers. Therefore, optimizing material handling (MH) time and costs aimed at reducing delivery errors, minimizing damage to goods and increasing customer service efficiency is directly conditioned by the automation of MH in logistics centers. Accordingly, this paper aims to rank and select smart MH solutions in logistics centers where deliveries are prepared for the supply of the city area. This paper proposes four smart solutions for a real company, and fourteen criteria are selected for the evaluation. A new hybrid Multi-Criteria Decision-Making model that combines the Fuzzy Analytic Hierarchy Process method, used to determine the criteria weights, and the Fuzzy COmprehensive distance-Based RAnking (FCOBRA) method, used to rank the alternatives, is proposed. The application of the model shows that the best alternative is the implementation of an autonomous forklift, which can greatly automate logistics activities and reduce the rate of delivery errors. The main contributions of this research are the definition of smart solutions, a framework for their evaluation and a new model for their ranking.

**Keywords:** smart material handling solutions; city logistics; industry 4.0; fuzzy AHP; fuzzy COBRA

## 1. Introduction

With increasingly pronounced globalization and the growth of the world population, urbanization and the volume of the flow of goods, interest in the optimization of logistics flows in cities, i.e., CL, is growing. CL includes all strategies, technologies and logistics solutions that support all participants and functions of urban spaces, regardless of their size, number, space and boundaries, following their individual and general interests and goals [1]. The growth of the urban population and the fact that about two-thirds of the total commodity flows begin or end in central city zones indicate the importance and problems of CL.

Customers' demands in the city are characterized by a high frequency, a wide assortment, home delivery, individual delivery, smaller deliveries, delivery at certain time intervals, etc. [1–6]. In addition, CL trends, such as low driving speeds, short routes, stochastic vehicle downtime, limited space, inadequate traffic infrastructure and many others, are permanently present in customer service in the city [7]. Moreover, increasing demand for home delivery and e-commerce (negatively) affects CL's performance, especially in the before and post COVID-19 era. Home delivery is an additional boost to the development of e-commerce because it enables purchased products to be delivered directly to the customer. Moreover, this creates an additional load for the delivery of goods because the number of deliveries increases [8].

Contemporary trends and the problems they generate in the field of CL have initiated research to find ways to eliminate them. Various CL solutions have been defined, which include the implementation of initiatives, concepts, measures and scenarios. The logistics center, as one of the CL subsystems and a key element of the success of most CL solutions [4,6,9], is of particular interest to researchers. Optimizing the MH process in logistics centers and improving its efficiency can improve the efficiency of the delivery of goods in the city and improve customer service. The process that greatly affects the performance and efficiency of logistics centers is MH, which refers to all types of cargo manipulation (storage, retrieval, transfer, transportation, etc.) within a rounded technological unit. Optimizing MH processes can eliminate the causes that lead to delivery delays, increased traffic jams, damage to goods, incorrect deliveries, etc., and can improve the entire process of the delivery of goods. In addition, by engaging appropriate MH technologies, work safety is increased, labor costs are reduced, the number of employees is reduced, etc. Following the above, this paper deals with the definition and valuation of smart MH solutions in logistics centers that are based on the engagement of modern MH equipment supported by Industry 4.0 (I 4.0) technologies. Smart solutions are observed in the context of overcoming key challenges in MH processes when preparing deliveries in the city area, using the example of a logistics company operating in Serbia. The analyzed company currently uses a fleet of forklifts for MH processes, but following the requirements of customers in the city, the company plans to replace the existing fleet with solutions based on modern MH equipment with integration with I 4.0 technologies. Accordingly, the goal of this paper is the ranking and selection of the most favorable MH solution that can lead to the improvement of the efficiency of the logistics center and, thus, to the improvement of the performance of the company's city logistics activities.

The evaluation of proposed alternative solutions based on modern MH equipment and I 4.0 technologies, concerning technical–technological, economic and normative criteria, is carried out. As it is a complex problem of multi-criteria decision making (MCDM), a new hybrid model combining the Fuzzy Analytic Hierarchy Process (FAHP) and Fuzzy COmprehensive distance-Based RAnking (FCOBRA) methods are defined for its solution, which is one of the main contributions of this paper. Defining smart solutions based on the engagement of Smart Automated Guided Vehicle (SAGV) forklifts, Autonomous Mobile Robots (AMRs), Goods-To-Person (GTP) and Unmanned Aerial Vehicles (UAV), as well as compatible I 4.0 technologies, represents another contribution of this paper. As the most favorable solution, the alternative based on implementing an SAGV forklift is obtained.

This paper is organized into several sections. After the introduction, a review of the literature in the domain of city logistics, problems with MH equipment selection, I 4.0 technologies and MCDM methods are presented. The third section provides the methodology that is used for the MH equipment selection. The fourth section provides insight into the analysis of alternative solutions for improving MH activities in logistics centers. In that context, this part also discusses and ranks the obtained results through sensitivity analysis. Sensitivity analysis is used to confirm if the achieved solution is optimal and to verify the stability of the developed model. The fifth section discusses the obtained results and the problem-solving framework, which summarizes the advantages and limitations of the applied model and the obtained results. In the fifth section, the practical and theoretical implications of the methodology and the obtained solution are also given. The last section provides concluding remarks and future research directions.

## 2. Literature Review

Due to the numerous challenges in CL, the efficient implementation of all processes in logistics centers, among which MH processes stand out, becomes one of the key elements of sustainable CL. Given the complexity of the observed problem, below is a literature review of the following key aspects: CL challenges, I 4.0 technologies and solutions that are applicable to MH activities, the selection of MH technologies and solutions, and an overview of MCDM methods used in this paper.

### 2.1. City Logistics Challenges

CL includes all strategies, technologies and logistics solutions that provide support to all participants and functions of the urban space, regardless of their size, number, space and boundaries, following their interests, limitations and goals [1] (Zečević and Tadić, 2006). The growth of the urban population as well as the fact that about two-thirds of the total flow of goods starts or ends in central city zones point to the importance of CL [7].

For this reason, in the last couple of decades, CL has been the subject of numerous research that has mainly dealt with identifying problems [4,5,10]; defining initiatives, concepts and measures to solve the problems [7,11–15]; defining the scenarios of the development of CL systems [9,16–19]; and developing and applying modern technologies in the field of CL [2,20–22].

The problems of CL arise as a result of the complex flow of goods caused by a large concentration of population in a relatively small area, a large number of urban functions, different business strategies, categories of generators and providers of logistics services. These flows have the characteristics of partiality, the spatial dispersion of generators, diversity regarding the structure of logistics chains, the frequency of a larger number of smaller deliveries, dynamism, stochasticity, etc. [1]. Problems are also a consequence of contemporary global trends, such as [18] inadequate decision making by city administrations [19], undefined policies and strategies for CL [19], an inadequate supply of logistics services [23], unfavorable locations of logistics centers [24], insufficiently developed logistics outsourcing [25], large participation of road transport in the realization of transport flows [26], low utilization of cargo space [27] and a negative impact on the environment [28].

To solve CL problems and make activities sustainable, large international projects were launched, and various CL initiatives were defined [5,11]. Some of the basic groups of initiatives are city administration initiatives (toll collection [28], permits and regulations [19] and parking and loading and unloading zones [29]), initiatives of logistics providers (co-operation of carriers [30], vehicle routing [31] and vehicle technology innovation [32]), infrastructure-related initiatives (logistics centers [16], underground logistics system [33], improvement of road infrastructure [33] and standardization of cargo units [1]) and initiatives related to the reorganization of logistics activities (transport exchanges [34] and intermodal transport [35]).

In addition to initiatives, scenarios that include a combination of several different initiatives, concepts and measures have also been defined to solve CL problems. Some of the authors who dealt with defining scenarios for solving logistics problems in certain cities are Van Duin et al. [36], Nathanail et al. [37] and Tadić et al. [6]. The defined scenarios refer to the way of performing, managing and controlling CL activities and flows in the territory of the city or individual city zones. In their paper, Van Duin et al. [36] proposed scenarios for solving CL problems based on a combination of CL initiatives and public urban transport to employ unused transport capacity. In their paper, Nathanail et al. [37] developed CL scenarios to reduce noise, traffic congestion and GHG emissions. Tadić et al. [6] selected CL scenarios for the central business zone. The scenarios were defined under the logistics concept of the city, taking into account all of the following stakeholders: citizens, users, logistics service providers and the city administration.

Additionally, modern I 4.0 technologies, used in the context of improving certain segments and activities of CL, stand out in the literature as one of the solutions to the problems of CL. Tadić et al. [38] analyzed the use of drones in city logistics. Taniguchi et al. [21] analyzed the use of IoT, Big Data, ITS, ICT and AI for the development of smart CL solutions. Sonneberg et al. [22] optimized the delivery of goods in stages using unmanned ground vehicles. They also selected the location for these vehicles to optimize the driving distance. In addition to the above, the key technologies that are applicable to logistics centers within CL are AGV [39], AS/RS [40], Pick by Voice [41], Pick by Light [42], electronic data exchange [42], virtual reality [43], etc.

Logistics centers stand out as one of the key elements in initiatives and scenarios for solving problems and increasing the sustainability of CL [4,5]. Various problems of logistics

centers' application in CL have been investigated in the literature so far. Nataraj et al. [44] combined the concept of flow consolidation in the logistics center and the cooperation strategy of city logistics participants to reduce the negative effects of CL. Pamučar et al. [45] optimized the delivery of goods from the logistics center by routing eco-vehicles. Crainic and Montreuil [46] proposed the concept of the synergy of logistics centers and the Physical Internet to increase the efficiency of the delivery of goods in the city. Rao et al. [47] selected the location of a logistics center with an emphasis on sustainability. However, the impact of MH activities on the performance of city logistics centers has not been analyzed in previous research. Accordingly, the research gap that this paper tries to cover is the improvement of the efficiency of the MH system in logistics centers through the selection of adequate smart solutions for MH.

*2.2. Technologies and Solutions of I 4.0 in MH*

Given the high importance of I 4.0 technologies in developing smart MH solutions, it is necessary to analyze previous research on the used technologies and, based on that, to propose potential alternative solutions for the analyzed case study. I 4.0 implies the use of computing, information and communication technologies, networks and real physical processes, thereby creating a Cyber–Physical System (CPS) [48–51]. The most frequently used I 4.0 technologies for improving the efficiency of logistics processes are IoT, the Industrial Internet of Things (IIoT), Cloud Computing (CC), robotics, AI, AR, Big Data (BG), Machine Learning (ML), 3D printing, Digital Twin (DT), Data Mining (DT), Process Mining (PM), AV, AGV, UAV, BC, etc. [49–51].

Krstić et al. [48] presented the possibility of the potential application of I 4.0 technologies in logistics. They considered the engagement of AV and AGV supported by IoT, AI, BC and other technologies of I 4.0. Mahdiraji et al. [52] examined the application of IoT, CC, BC and simulations of all processes of the pharmaceutical industry, including raw material procurement, manufacturing, distribution and reverse logistics. Sharma et al. [53] investigated the impact of the LARG (Lean–Agile–Resilience–Green) paradigm, supported by I 4.0 technologies, on management processes and SC performance. Tadić et al. [38] analyzed the use of drones in CL. Tadić et al. [20] proposed smart, sustainable CL solutions as a combination of various CL initiatives, measures, concepts and I 4.0 technologies (AV, UAV and robotics). They aimed to mitigate GHG emissions, urban congestion and other effects of unsustainable logistics.

Tadić et al. [2] used I 4.0 technologies for defining CL concepts based on UAV, AV and AVG supported by IoT, BC, CC and others. Krstić et al. [54] applied I 4.0 solutions and technologies in the fields of logistics and circular economy, aiming to increase the efficiency of the reverse logistics sector in the food industry. For this purpose, they considered the engagement of AV and AGV supported by IoT, 3D, AI, BC and others. Ghadge et al. [55] highlighted the importance of I 4.0 in the automotive industry. They stated that the use of IoT, BC and CPS can contribute to the improvement of performance in the segment of forward and reverse logistics within the framework of a green supply chain (SC).

In the reviewed literature, an absence of consideration of the application of I 4.0 technologies in the creation of solutions for smart MH in logistics centers is identified. Therefore, this paper aims to provide adequate guidelines for defining and selecting an MH alternative for improving the efficiency of customer service in the city.

*2.3. MH Equipment Selection*

The problem of selecting MH equipment is widespread, and it concerns almost all subsystems of logistics. These problems address transport and handling technology, storage technology, sorting technology, order-picking technology, etc. In this paper, the emphasis is on the selection of transport and handling technology in logistics centers. Complex MH activities are performed in logistics centers using various equipment (forklifts, cranes, AGVs, roller and belt conveyors, etc.). Nguyen et al. [56] and Sathyam et al. [57] dealt with the selection of the appropriate types of conveyor technologies for performing MH activities.

Mathev and Sahu [58] addressed a couple of problems when selecting MH equipment. The first problem was related to the selection of the conveyor, whereas the second was focused on the selection of the AGV. Fazlollahtabar et al. [59] and Satoglu and Türkekul [60] solved the problem of selecting a forklift in a warehouse by considering manual forklifts, manual hydraulic and hydraulic forklifts, semi-electric forklifts and electric forklifts. Zubair et al. [61] estimated the efficiency of the engagement of AGVs, forklifts and belt conveyors in a warehouse. Goswami et al. [62] focused their research on optimizing the selection of MH assets by analyzing conveyors, AGVs and AMRs.

In previous research dealing with MH equipment selection, authors have mostly considered typical technologies, which include various types of forklifts, rollers and belt conveyors, AGVs, etc. The employment of smart solutions and technology I 4.0 has not been sufficiently explored. As the engagement of I 4.0 technologies enables greater savings, increased productivity of the system, greater safety for employees and more promising use of resources, this paper focuses on selecting appropriate combinations of I 4.0 technologies and MH equipment in logistics centers.

### 2.4. Review of MCDM Methods in MH Equipment Selection

To determine the essential advantages and prominent disadvantages of potentially applicable solutions for optimizing MH activities in logistic centers, in this research, some of the innovative MCDM methods can be used while admiring the adequate criteria. Different methods and tools can be used to solve the MH equipment selection problem. In the literature, authors have used heuristics and metaheuristics, simulations, mathematical programming, Multi-Objective Decision Making (MADM) and MCDM, etc. In recent years, MCDM methods have gained more importance in solving the MH equipment selection problem. Numerous methods are used, such as AHP, the Technique for Order of Preference by Similarity to Ideal Solution (TOPSIS), (srb.) VIšekriterijumska optimizacija I KOmpromisno Rešenje (VIKOR), Simple Additive Weighting (SAW), the Preference Ranking Organization METHod for Enrichment of Evaluations (PROMETHEE), ELimination and Choice Expressing REality (ELECTRA), Evaluation Based On Distance From Average Solution (EDAS), Combinative Distance-based ASsesment (CODAS), Multi-Objective Optimization on the basis of Ratio Analysis (MOORA) and many others, as well as a combination of these methods or developing these methods in a fuzzy and gray environment.

The AHP method is theoretically easy to understand and simple for practical application [63]; it allows for the consideration of both quantitative and qualitative parameters [64] and for the comparison of pairs of elements, on which it is based, making it one of the most transparent and technically appropriate methods for determining the weights (importance) of the problem elements [65]. The AHP method requires greater effort to obtain a solution compared to some other methods that perform a direct ranking of elements, such as the Simple Multi-Attribute Rating Technique (SMART) and Stepwise Weight Assessment Ratio Analysis (SWARA) methods [66], but it also allows decision makers to better understand the relative importance and interactions between elements and perform a more precise evaluation, which results in better-quality solutions [67,68]. In addition, compared to methods that require complete consistency, e.g., the Measuring Attractiveness by a Categorical-Based Evaluation Technique (MACBETH) or Full Consistency Method (FUCOM), the AHP method is flexible and allows increasing or decreasing the tolerance threshold according to the preferences of decision makers [67,69]. Because of all of the above, AHP is one of the most commonly used MCDM methods, especially for determining the weight of criteria. Independently or in combination with other methods, it has found wide applications for solving various problems. Horňákováet al. [70] applied the AHP method for the selection of MH equipment, including forklifts, AGV forklifts and AMRs. Parameshwaran et al. [71] ranked and selected robotic technologies. Based on the attitudes of decision makers, relevant criteria were selected using Fuzzy Delphi, and the FAHP method was used to determine the relative weights of the criteria. FVIKOR and FTOPSIS were used to rank the alternatives. Nguyen et al. [56] used FAHP and Fuzzy Additive Ratio Assessment (FARAS) for ranking and selecting conveyors.

FAHP was used to determine the relative weights of the criteria, and FARAS was used to rank the alternatives. Zubair et al. [61] used the AHP method to select MH equipment in a warehouse. Satoglu and Türkekul [60] used the AHP method to determine the relative weights of the criteria, after which they ranked the alternatives using the MOORA method. The mentioned advantages of the method, as well as the fact that the hierarchical structure of the AHP method corresponds to the structure of the problem that is considered in this paper, are the key reasons for choosing this method. The fuzzy extension of the AHP method is used because the evaluations are made by decision makers whose cognitive processes are better modeled by using fuzzy logic.

The main advantage of the COBRA method over other distance-based MCDM methods is its comprehensiveness. The alternatives are ranked according to the comprehensive distances of each alternative from the three types of solutions, namely positive ideal, negative ideal and average solutions, unlike the other distance-based methods, which rank the alternatives using two such solutions at most. In addition, COBRA uses both Euclidian and taxicab distance measurements to calculate distances for all of these solutions, which ensures higher reliability for the obtained solution. Two types of distance measurements also allow for a fine differentiation between the distance values, which could otherwise be close and thus insufficiently reliable for making a decision. Although very young, this method has already found applications for solving various problems. Krstić et al. [72] combined COBRA and AHP methods to evaluate the applicability of I 4.0 technologies in reverse logistics in the agri-food sector. The AHP method was used to determine the weights of the criteria, and the COBRA method was used to obtain the ranking of the alternatives. Popović et al. [73] used a combination of the MEREC (Method based on the effects of criteria removal) and COBRA methods for selecting an adequate strategy for the development of e-commerce. The MEREC method is applied to define the weight of the criteria, and the COBRA method is used for the evaluation and ranking of the considered alternatives. Popović et al. [74] selected a provider of RFID tags using a combination of PIPRECIA-S (Simplified PIvot Pairwise RElative Criteria Importance Assessment), PSI (objective Preference Selection Index) and the COBRA method. Krstić et al. [75] used a combination of fuzzy Delphi, Fuzzy Analytic Network Process (FANP) and FCOBRA methods for evaluating and ranking I 4.0 technologies based on the aspect of thedevelopment and improvement of sustainable circular economy systems and circular SC. Fuzzy Delphi and FANP were used to determine the weights of the criteria, and FCOBRA was used to determine the ranking. Verma et al. [76] proposed an integrated MCDM method to evaluate suppliers in the steel manufacturing sector, which includes (Best–Worst Method) BWM and COBRA. In this method, the weights of criteria are determined via BWM, and then suppliers are ranked using COBRA. The COBRA method is chosen in this paper for the alternatives' ranking due to the mentioned advantages and proven applicability. The fuzzy version of the method is used for the same reasons as the fuzzy AHP method. A review of the previous research shows that, so far, no combination of FAHP and FCOBRA methods has been performed for selecting MH equipment or any other problem, which is another research gap that this paper tries to cover.

### 3. Hybrid MCDM Model

In this research, a new hybrid MCDM model is created and used for ranking and selecting MH equipment. The hybrid model combines the FAHP [77] and FCOBRA methods [75]. FAHP is used to determine the relative weights of the criteria, and FCOBRA is used to rank the alternatives. Methods developed in a fuzzy environment are suitable for describing insufficiently precise data, interval grades, etc. Below is a step-by-step description of the method.

**Step 1**: Define the problem structure.

The problem needs to be structured hierarchically in accordance with the AHP and FAHP method. The structure includes defining the goal, criteria, sub-criteria and alternatives.

**Step 2**: Pair-wise comparisons.

For the pair-wise comparison procedure in the AHP method, Satie's scale (1–9) is used [64]. According to FAHP, it is necessary to define fuzzy sets on the same scale. Pair-wise comparisons need to be made for all sub-criteria and criteria concerning a higher level of the hierarchy. The linguistic ratings and the corresponding triangular fuzzy number that are utilized to rate the elements are presented in Table 1.

**Table 1.** Fuzzy scale for the comparison of criteria/alternatives.

| Linguistic Term | Fuzzy Scales |
|---|---|
| Absolutely preferable/better (AP/B) | (8,9,10) |
| Very preferable/better (VP/B) | (7,8,9) |
| Strongly preferable/better (SP/B) | (6,7,8) |
| Pretty preferable/better (PP/B) | (5,6,7) |
| Quite preferable/better (QP/B) | (4,5,6) |
| Moderately preferable/better (MP/B) | (3,4,5) |
| Remotely preferable/better (RP/B) | (2,3,4) |
| Barely preferable/better (BP/B) | (1,2,3) |
| Equally important/good (EI/G) | (1,1,2) |

**Step 3**: Formation of the fuzzy matrix $\widetilde{\in}$

In this step, the fuzzy matrix that is used in the FAHP method for pair-wise comparison is defined. A matrix is formed for each set of criteria or sub-criteria that are compared with each other.

$$\widetilde{\in} = \begin{bmatrix} \widetilde{a}_{11} & \cdots & \widetilde{a}_{in} \\ \vdots & \ddots & \vdots \\ \widetilde{a}_{n1} & \cdots & \widetilde{a}_{nn} \end{bmatrix}, \tag{1}$$

**Step 4**: Determining the relative weight of the criteria.

A priority vector is computed for each comparison W; $W = (w_1, \ldots, w_n) > 0$, $\sum_{j=1}^{n} w_j = 1$. The value of vector W in the FAHP method can be calculated using various techniques and methods. For the purposes of this paper, the "Logarithmic Fuzzy Preference Programming" (LFPP) method [78] is chosen. Each triangular fuzzy number is defined as follows: $\widetilde{a}_{ij} = (l_{ij}, m_{ij}, u_{ij})$. The LFPP method is based on the calculation of the logarithmic function of the fuzzy number, as follows:

$$\ln \widetilde{a}_{ij} \approx (\ln l_{ij}, \ln m_{ij}, \ln u_{ij}); i, j = 1, \ldots, n, \tag{2}$$

$$MinJ = (1 - \lambda)^2 + M \times \sum_{i=1}^{n-1} \sum_{j=i+1}^{n} \left( \delta_{ij}^2 + \eta_{ij}^2 \right), \tag{3}$$

$$s.t. \begin{cases} x_i - x_j - \lambda \ln \left( m_{ij}/l_{ij} \right) + \delta_{ij} \geq \ln l_{ij}, i = 1, \ldots, n-1; j = i+1, \ldots, n \\ -x_i + x_j - \lambda \ln \left( u_{ij}/m_{ij} \right) + \eta_{ij} \geq -\ln u_{ij}, i = 1, \ldots, n-1; j = i+1, \ldots, n \\ \lambda, x_i \geq 0, i = 1, \ldots, n \\ \delta_{ij}, \eta_{ij} \geq 0, i = 1, \ldots, n-1; j = i+1, \ldots, n \end{cases}, \tag{4}$$

where

$x_i^*(i = 1, \ldots, n)$ is the optimal solution, and $M = 10^3$ is a specified sufficiently large number.

Non-negative variables $\delta_{ij}$ and $\eta_{ij}$, $i = 1, \ldots, n-1; j = i+1, \ldots, n$ are defined to prevent the membership degree $\lambda$ from taking a negative value in order to fulfill the following inequalities:

$$\ln w_i - \ln w_j - \lambda \ln \left( \frac{m_{ij}}{l_{ij}} \right) + \delta_{ij} \geq \ln l_{ij}, i = 1, \ldots, n-1; j = i+1, \ldots, n, \tag{5}$$

$$-\ln w_i + \ln w_j - \lambda \ln\left(m_{ij}/l_{ij}\right) + \eta_{ij} \geq -\ln u_{ij}, i = 1, \ldots, n-1; j = i+1, \ldots, n, \quad (6)$$

The crisp normalized priority vector of the matrix $\widetilde{A} = \left(\widetilde{a}_{ij}\right)_{n \times m}$ can be obtained as follows:

$$W_i^* = \frac{exp\left(x_i^*\right)}{\sum_{j=1}^{n} exp\left(x_j^*\right)}, i = 1, \ldots n, \quad (7)$$

where

$$exp(x_i^*) = e^{x_i^*}, \quad (8)$$

The stability of the results is controlled by calculating the Consistency Ratio (CR) for each matrix [31]:

$$CR = \frac{CI}{RI}, \quad (9)$$

where the Consistency Index (CI) is calculated as

$$CI = \frac{Z_{max} - 0}{0 - 1}, \quad (10)$$

The Random Index (RI) depends on the matrix size and is given in Saaty [64]. $Z_{max}$ in Equation (9) stands for the principal eigenvalue of the matrix $\widetilde{\in}$. CR values need to be less than 0.10 for all comparisons.

**Step 5:** Formation of the fuzzy matrix $\widetilde{F}$

In the second part of the model, FCOBRA is used to rank the alternatives. After obtaining the relative weights using the FAHP method, it is necessary to define the input parameters for the FCOBRA method [30]. A matrix is formed by comparing alternatives against all criteria.

$$\widetilde{F} = \left[ \begin{pmatrix} \widetilde{f}_{11} & \cdots & \widetilde{f}_{1o} \\ \vdots & \ddots & \vdots \\ \widetilde{f}_{p1} & \cdots & \widetilde{f}_{po} \end{pmatrix} \right], \quad (11)$$

where

$\widetilde{f}_{kj} = \left(l_{kj}, m_{kj}, u_{kj}\right)$ are the evaluations of the alternatives $k(i = 1, \ldots, p)$ in relation to the criteria $j(j = 1, \ldots, o)$ that are obtained using the scale given in Table 1; $p$ is the total number of the alternatives taken into consideration; $o$ is the total number of criteria; and $l_{kj}, m_{kj}, u_{kj}$ are the lower, middle and upper values of the triangular fuzzy number $\widetilde{f}_{kj}$, respectively.

**Step 6**: Normalization of the fuzzy matrix.

The normalized fuzzy matrix is obtained as follows:

$$\widetilde{\Phi} = \left[\widetilde{\varphi}_{kj}\right]_{p \times o}, \quad (12)$$

where

$\widetilde{\varphi}_{kj} = \left(\alpha_{kj}, \theta_{kj}, \rho_{kj}\right)$ is the normalization triangular fuzzy number, whose values are obtained as follows:

$$\alpha_{kj} = \frac{l_{kj}}{\left(\max_k u_{kj}\right)}, \forall k = 1, \ldots, p; \forall j = 1, \ldots, o\text{—lower values fuzzy number,} \quad (13)$$

$$\theta_{kj} = \frac{m_{kj}}{\left(\max_k u_{kj}\right)}, \forall k = 1, \ldots, p; \forall j = 1, \ldots, o\text{—middle values fuzzy number,} \quad (14)$$

$$\rho_{kj} = \frac{u_{kj}}{\left(\max_k u_{kj}\right)}, \forall k = 1, \ldots, p; \forall j = 1, \ldots, o\text{—upper values fuzzy number,} \tag{15}$$

**Step 7**: Weighted normalized fuzzy decision matrix $\widetilde{\Phi}_w$

After normalization, by multiplying the matrix with the relative weight of the criteria ($w_j$) that are obtained via the FAHP method, a weighted fuzzy matrix is obtained, and its mathematical notation is as follows:

$$\widetilde{\Phi}_w = \left[w_j \times \widetilde{\varphi}_{kj}\right]_{p \times o}, \tag{16}$$

**Step 8**: Determination of fuzzy ideal, fuzzy anti-ideal and fuzzy average solutions.

For each criterion function, the fuzzy ideal ($\widetilde{PIS}_j$), fuzzy anti-ideal ($\widetilde{NIS}_j$) and fuzzy average solution ($\widetilde{AS}_j$) are determined as follows:

$$\widetilde{PIS}_j = \left(\alpha_{kj}^{PIS}, \theta_{kj}^{PIS}, \rho_{kj}^{PIS}\right) = \left(\max_k\left(w_j \times \alpha_{kj}\right), \max_k\left(w_j \times \theta_{kj}\right), \max_k\left(w_j \times \rho_{kj}\right)\right), \forall_j = 1, \ldots, o \text{ for } j \in J^B, \tag{17}$$

$$\widetilde{PIS}_j = \left(\alpha_{kj}^{PIS}, \theta_{kj}^{PIS}, \rho_{kj}^{PIS}\right) = \left(\min_k\left(w_j \times \alpha_{kj}\right), \min_k\left(w_j \times \theta_{kj}\right), \min_k\left(w_j \times \rho_{kj}\right)\right), \forall_j = 1, \ldots, o \text{ for } j \in J^C, \tag{18}$$

$$\widetilde{NIS}_j = \left(\alpha_{kj}^{PIS}, \theta_{kj}^{PIS}, \rho_{kj}^{PIS}\right) = \left(\min_k\left(w_j \times \alpha_{kj}\right), \min_k\left(w_j \times \theta_{kj}\right), \min_k\left(w_j \times \rho_{kj}\right)\right), \forall_j = 1, \ldots, o \text{ for } j \in J^B, \tag{19}$$

$$\widetilde{NIS}_j = \left(\alpha_{kj}^{PIS}, \theta_{kj}^{PIS}, \rho_{kj}^{PIS}\right) = \left(\max_k\left(w_j \times \alpha_{kj}\right), \max_k\left(w_j \times \theta_{kj}\right), \max_k\left(w_j \times \rho_{kj}\right)\right), \forall_j = 1, \ldots, o \text{ for } j \in J^C, \tag{20}$$

$$\widetilde{AS}_j = \left(\alpha_{kj}^{AS}, \theta_{kj}^{AS}, \rho_{kj}^{AS}\right) = \left(\text{mean}_k\left(w_j \times \alpha_{kj}\right), \text{mean}_k\left(w_j \times \theta_{kj}\right), \text{mean}_k\left(w_j \times \rho_{kj}\right)\right), \forall_j = 1, \ldots, o \text{ for } j \in J^B, J^B, \tag{21}$$

**Step 9**: Determining the distance of the alternative from the fuzzy ideal, fuzzy anti-ideal and fuzzy average solutions.

For each alternative, it is necessary to determine the distance of the alternative from the fuzzy ideal $\left(d\left(\widetilde{PIS}_j\right)\right)$, fuzzy anti-ideal $\left(d\left(\widetilde{NIS}_j\right)\right)$, fuzzy positive average $\left(d\left(\widetilde{AS}_j\right)^+\right)$ and fuzzy negative average $\left(d\left(\widetilde{AS}_j\right)^-\right)$ solutions, as follows:

$$d\left(\widetilde{S}_j\right) = dE\left(\widetilde{S}_j\right) + \xi \times dE\left(\widetilde{S}_j\right) \times dT\left(\widetilde{S}_j\right) \forall_j = 1, \ldots, o, \tag{22}$$

where

$\widetilde{S}_j$ is any solution $\left(\widetilde{PIS}_j, \widetilde{NIS}_j, or \widetilde{AS}_j\right)$, and $\xi$ is the correction coefficient obtained as follows:

$$\xi = \max_k dE\left(\widetilde{S}_j\right)_k - \min_k dE\left(\widetilde{S}_j\right)_k, \tag{23}$$

$dE\left(\widetilde{S}_j\right)$ i and $dT\left(\widetilde{S}_j\right)$ denote the Euclidian and taxicab distances, respectively, which are for the positive ideal solution, obtained as follows:

$$dE\left(\widetilde{PIS}_j\right)_k =$$

$$\sum_{j=1}^{o} \sqrt{\left(\left(\alpha_{kj}^{PIS} - w_j \times \alpha_{kj}\right)^2 + 4 \times \left(\theta_{kj}^{PIS} - w_j \times \theta_{kj}\right)^2 + \left(\rho_{kj}^{PIS} - w_j \times \rho_{kj}\right)^2\right)/6}, \forall k = \tag{24}$$

$$1,\ldots,p; \forall j = 1,\ldots,o,$$

$$dT\left(\widetilde{PIS}_j\right)_k =$$

$$\sum_{j=1}^{o} \sqrt{\left(\left|\alpha_{kj}^{PIS} - w_j \times \rho_{kj}\right|^2 + 4 \times \left|\theta_{kj}^{PIS} - w_j \times \theta_{kj}\right|^2 + \left|\rho_{kj}^{PIS} - w_j \times \alpha_{kj}\right|^2\right)/6}, \forall k = \tag{25}$$

$$1,\ldots,p; \forall j = 1,\ldots,o,$$

The anti-ideal solution is obtained as follows:

$$dE\left(\widetilde{NIS}_j\right)_k =$$

$$\sum_{j=1}^{o} \sqrt{\left(\left(\alpha_{kj}^{NIS} - w_j \times \alpha_{kj}\right)^2 + 4 \times \left(\theta_{kj}^{NIS} - w_j \times \theta_{kj}\right)^2 + \left(\rho_{kj}^{NIS} - w_j \times \rho_{kj}\right)^2\right)/6}, \forall k = \tag{26}$$

$$1,\ldots,p; \forall j = 1,\ldots,o,$$

$$dT\left(\widetilde{NIS}_j\right)_k =$$

$$\sum_{j=1}^{o} \sqrt{\left(\left|\alpha_{kj}^{NIS} - w_j \times \rho_{kj}\right|^2 + 4 \times \left|\theta_{kj}^{NIS} - w_j \times \theta_{kj}\right|^2 + \left|\rho_{kj}^{NIS} - w_j \times \alpha_{kj}\right|^2\right)/6}, \forall k = \tag{27}$$

$$1,\ldots,p; \forall j = 1,\ldots,o,$$

The positive distance from the average solution is obtained as follows:

$$dE\left(\widetilde{AS}_j\right)_k^+ =$$

$$\sum_{j=1}^{o} \sqrt{\left(\tau^+ \left(\alpha_{kj}^{AS} - w_j \times \alpha_{kj}\right)^2 + 4 \times \tau^+ \left(\theta_{kj}^{AS} - w_j \times \theta_{kj}\right)^2 + \tau^+ \left(\rho_{kj}^{AS} - w_j \times \rho_{kj}\right)^2\right)/6}, \forall k \tag{28}$$

$$= 1,\ldots,p; \forall j = 1,\ldots,o,$$

$$dT\left(\widetilde{AS}_j\right)_k^+ =$$

$$\sum_{j=1}^{o} \sqrt{\left(\tau^+ \left|\alpha_{kj}^{AS} - w_j \times \rho_{kj}\right|^2 + 4 \times \tau^+ \left|\theta_{kj}^{AS} - w_j \times \theta_{kj}\right|^2 + \tau^+ \left|\rho_{kj}^{AS} - w_j \times \alpha_{kj}\right|^2\right)/6}, \forall k \tag{29}$$

$$= 1,\ldots,p; \forall j = 1,\ldots,o,$$

where

$$\tau^+ = \begin{cases} 1 \text{ if } \widetilde{AS} < w_j \times \varphi_{kj} \\ 0 \text{ if } \widetilde{AS} > w_j \times \varphi_{kj} \end{cases}, \tag{30}$$

The negative distance from the average solution is obtained as follows:

$$dE\left(\widetilde{AS}_j\right)_k^- = \sum_{j=1}^{o} \sqrt{\left(\tau^-\left(\alpha_{kj}^{AS} - w_j \times \alpha_{kj}\right)^2 + 4 \times \tau^-\left(\theta_{kj}^{AS} - w_j \times \theta_{kj}\right)^2 + \tau^-\left(\rho_{kj}^{AS} - w_j \times \rho_{kj}\right)^2\right)/6}, \forall k \tag{31}$$
$$= 1, \ldots, p; \forall j = 1, \ldots, o,$$

$$dT\left(\widetilde{AS}_j\right)_k^- = \sum_{j=1}^{o} \sqrt{\left(\tau^-\left|\alpha_{kj}^{AS} - w_j \times \rho_{kj}\right|^2 + 4 \times \tau^-\left|\theta_{kj}^{AS} - w_j \times \theta_{kj}\right|^2 + \tau^-\left|\rho_{kj}^{AS} - w_j \times \alpha_{kj}\right|^2\right)/6}, \forall k \tag{32}$$
$$= 1, \ldots, p; \forall j = 1, \ldots, o,$$

where

$$\tau^- = \begin{cases} 1 \; if \; \widetilde{AS} < w_j \times \varphi_{kj} \\ 0 \; if \; \widetilde{AS} > w_j \times \varphi_{kj} \end{cases}, \tag{33}$$

**Step 10**: Ranking of alternatives.

The alternatives are ranked according to the increasing values of the comprehensive distances $dC_k$, obtained as follows:

$$dC_k = \frac{d\left(\widetilde{PIS}_j\right)_k - d\left(\widetilde{NIS}_j\right)_k - d\left(\widetilde{AS}_j\right)_k^+ - d\left(\widetilde{AS}_j\right)_k^-}{4}, \forall k = 1, \ldots, p, \tag{34}$$

## 4. Problem Statement

Due to increased urbanization, online ordering and e-commerce, logistics companies face numerous challenges in meeting customer demands in the city. The application of new strategies and business models, trends of increasing frequency and decreasing the delivery size additionally affect the complexity of the challenges in CL. In the processes of preparing goods for delivery in logistics centers, MH processes play one of the key roles. Optimizing the MH process can contribute to overcoming various challenges in CL.

Accordingly, this paper aims to rank and select smart solutions for MH activities for a logistics center of a leading German logistics company in the Serbian market. Solutions are based on combinations of modern MH equipment and I 4.0 technologies. The main activity of the logistics company is the preparation and delivery of consumers' goods for the City of Belgrade. The investigated company uses a fleet of electric forklifts for MH activities and the preparation of goods for delivery. It was estimated that, by engaging smart MH solutions, the company can respond more efficiently to customers' requirements and improve its competitive position in a vulnerable market. Customers' requirements in the city are characterized by frequent deliveries, small quantities, pre-defined time windows and a variable assortment of goods. Limitations, such as traffic jams, delays, insufficient utilization of transport capacities and negative environmental impacts, create additional motivation for engaging smart MH technologies in the company's logistics center. Although there are numerous challenges in the logistics processes of the company, some of the key ones concerning MH activities in logistics centers are as follows:

- Damage to goods during handling (storage, retrieval, internal transport, loading, etc.), which leads to delivery delays;
- Frequent employee errors in order picking activities that lead to wrong deliveries;
- Congestion of the transshipment dock, and waiting for vehicles to load/unload leads to delivery delays;
- A wrong storage policy leads to an extension of the order-picking time and delivery delays;
- Crossing the intralogistics flows causes an extension of MH technologies' operating cycles.

Due to the existence of time windows in which it is possible to deliver goods to the city, any delay in the execution of logistics activities in a company has a negative impact on customer service. Incorrectly and inefficiently made deliveries lead to direct customer dissatisfaction. Solutions based on modern MH equipment and I 4.0 technologies provide

high efficiency, low operating costs, high work safety, less possibility of error and damage to goods, etc., which create conditions for sustainable business for the company.

### 4.1. Defining Alternative Smart Solutions

This paper proposes (smart) solutions based on modern MH equipment, such as SAGV forklifts, AMRs, GTP and UAVs, as well as I 4.0 technologies, such as the Industrial Internet of Things (IIoT), Global Positioning System supported by CC-Cloud GPS (CGPS), RFID, BD, DM, AI and AR with integrated Decision Support Systems (DSS) and Warehouse Management Systems supported by CC-Cloud WMS (CWMS). With IIoT sensors [79,80] and RFID tags [81,82], communication is carried out between every device and element in the warehouse. IIoT sensors record and update information on warehouse operations. The routing and navigational functions of smart MH equipment depend on CGPS [83,84]. The system that integrates and monitors all warehouse operations is known as CWMS [85,86]. Data regarding every process in the warehouse is gathered, examined, and processed using BD and DM technologies [87,88]. With the combination of AI and AR, the DSS is a system that offers several bases that make it simpler for managers to make decisions [89].

*SAGV forklift (A1)*. AGVs have widespread use in MH operations within the automation of storage technologies. One of the categories of AGVs is the AGV forklift, which is considered the most commonly used equipment for transport and MH operations in modern smart systems [90]. Unlike the traditional driverless AGV forklift, which is controlled by a central computer and generally has a fixed route, the Smart AGV (SAGV) forklift includes its own computer, providing logistics operations with more flexibility and independence from other forklifts in the fleet [90–92]. The mobility of smart AGVs is possible with the support of CGPS and RFID technology. CGPS technology enables the management of MH equipment in warehouses (this technology provides data on the position of MH equipment in real-time) [93,94]. Moreover, RFID tags are used to indicate the appropriate location where the next warehouse assignment should be accepted. The SAGV forklift communicates with the control system via a Wi-Fi network that connects all RFID and IIoT sensors [79–81]. SAGV forklifts have safety scanners that operate based on AR, which prevent them from encountering barriers such as employees, equipment, goods, etc., thus providing high-level work safety [88,91–95]. CWMS is necessary for the communication of all connected entities (MH and other equipment, goods, etc.) with the central computer. The primary functions that a CWMS should provide are information on inventory status, the planning and scheduling of an assignment to MH equipment, the arrival time and position of external transport vehicles, the execution of an assignment and order status [84]. In the system, all MH equipment and other equipment has an IIoT sensor that collects and saves data about their current state and potential damage due to collisions. In autonomous technology, it is required to protect and process a considerable amount of data, which can be performed with the support of BD and DM [86,87]. BD helps collect and save data on routes, performed warehouse assignments, serviced warehouse locations, the time required to perform each warehouse assignment, information on the times of orders' fulfillment, the level of errors during order picking, the degree of damage to an assignment and order status. The collection and processing of these data are provided using DM. The processed data from AI technology in integration with DSS is used as input data for determining the optimal route, the order of performing warehouse assignments, storage and retrieval strategies, etc. Considering the foregoing, SAGV forklifts have the imperative to become one of the main automation technologies in logistics centers oriented toward improving customer service in the city [90,91,94].

The advantages of SAGV forklifts are their high positioning accuracy, significant lifting height and functionality in narrow corridors, resulting in increased utilization of storage area by 30–40% compared to traditional electric forklifts and in the increased efficiency of customer service [91–94]. The social sustainability of SAGVs is mainly related to the humanization of MH. Reducing labor costs, improving inventory management and reducing employee errors are just some of the key elements of the economic advantages

of this alternative. Environmental sustainability is primarily manifested through the use of electricity for propulsion, which significantly contributes to reductions in greenhouse gas (GHG) emissions, including a low noise level in work environments. Some of the key disadvantages are relatively high investment costs, increased maintenance costs, additional employee training requirements, etc. [92–94].

*AMR (A2).* AMRs symbolize a fully automated system that can independently complete intralogistics activities because it uses onboard sensors and processors for automatic MH without the need for physical controls [96]. Routing in the AMR system is accomplished using CGPS, which traces the asset's location in real time and provides coordinates for further movement [97]. In contrast to traditional AGV systems with a fixed route, AMRs have much greater flexibility because of AI and AR support [89]. With the support of AI and AR, AMRs "learn" their environment by registering its own location and then dynamically plans routes based on current environmental conditions and logistics requirements [88,89]. When barriers arise on a defined route, AI technology as a base component of AMRs at the same time reorganizes and re-optimizes the route to the next point [98,99]. BD collects data on completed assignments and routes as a base for DM to optimize the route to the next warehouse assignment [86,87]. Because AMRs do not have the possibility of active pick-up and disposal of material, IIoT technology provides adequate information about the engagement of pick-up mechanisms, such as robotic arms, as well as collaboration with other equipment, such as cranes [78,79,97–100]. Thus, upon completion of the warehouse task, the CWMS data provided by the BD is forwarded to the auxiliary equipment, which, at that moment, engages in the next MH position [84]. As with SAGVs, the basic task of CWMS is to provide all the necessary information that is a prerequisite for performing the warehouse assignment, which is related to the position of the external transportation vehicle, its start and end times in MH activities, the contents of deliveries, and the process of performing intralogistics activities [97]. AMRs are a contemporary technology that provides additional benefits in implementation due to its simple integration into the existing storage system and effective engagement when other technologies fail in MH activities [98,100].

The main advantages of AMRs are high flexibility in material transport, the possibility of autonomous decision making, reductions in crossing flows, etc. [99–101]. Lowering the cost of logistics activities, such as maintaining low operating and maintenance costs, contributes to economic sustainability. Reductions in employees engaged in MH activities significantly increases work safety, which also makes this alternative socially sustainable. Environmental sustainability is reflected in the use of electric energy for propulsion, which completely eliminates GHG emissions and produces low noise levels. However, AMRs have challenges in terms of management, IT support and updating data. One of the main disadvantages is not having the ability to actively load freight. As a result, robotic arms, cranes or other devices that can provide pick-up of the load need to be used on transshipment decks [55,56]. Lower capacity and speeds, including longer battery charging times for AMRs compared to AGV forklifts, are some of the additional limitations that require consideration when employing them for MH activities [97–99].

*GTP (A3).* GTP represents an automated system that integrates an AMR and an Automated Storage and Retrieval System (AS/RS) to increase the efficiency of MH activities in logistics centers. In some ways, this technology is an upgrade to AS/RS, which, in addition to automatic storage and retrieval, also performs order-picking activities [102,103]. By applying CWMS, goods are stored in predefined locations, from which they are picked up and automatically transported directly to the picker, eliminating unnecessary movement time and providing accurate stock data [84,102]. Consequently, using AMRs increases the efficiency of employees, and AS/RS has high warehouse density and commodity flow. In this contemporary warehouse system, racks and/or shelves are used to hold inventory and are also in a high-density arrangement on the floor. According to MH requirements, the AMR moves under the rack and raises it to a position at the appropriate place. With the help of IIoT shelf sensors and AMRs, safe movement within the logistics center is ensured [79,80]. BD technology saves and stores data from CWMS and sends it in the appropriate format

to DM technology for processing. The data refer to the sequence of order execution, the number of requested goods, the direction of delivery as well as other relevant information that is necessary for the efficient execution of the order [85–87]. These data can also be used when defining the order of rack positioning. The pick-up–delivery station, as one of the most responsible entities in MH, can have a classic configuration with only a monitor and scanner, but it can also be supported by Pick by Vision technology and scales, and can be equipped with packaging material and other equipment for finalizing warehouse assignment. The mentioned and other demanding tasks can be effectively performed by engaging AI and AR technologies, thus providing employees with timely information about the goods they handle [99,102–104]. In this way, repetitive and frequent activities are minimized to the point of elimination, and the abilities and skills of the employees are directed toward the improvement of the MH process, thus contributing to more efficient customer service [103,104].

The GTP system is designed to reduce the workload of employees, provide high storage density and improve inventory flow. The system's ability to handle a wide range of goods makes it suitable for the reallocation of orders, easy replenishment of inventories, and other logistical activities. The GTP system automatically locates and stores goods to increase throughput and utilization, increases precision and enables delivery on the same day as the order is received, which is especially significant in the city [102,104]. Using GTP increases accuracy and, at the same time, eliminates errors during ordering, which is extremely important in systems in which there is a high daily demand (e-commerce). Reducing the costs of logistics activities in the GTP system contributes to economic sustainability. The elimination of human labor significantly increases work safety, which makes this alternative socially sustainable. Environmental sustainability is reflected in the use of electric energy for propulsion (thus eliminating GHG emissions) and low noise levels [104,105]. Using a CC-hosted WMS, GTP can be efficiently controlled to automate the picking process, bringing goods directly to the operator. The main disadvantages of this technology are high implementation costs, less flexibility due to a fixed number of storage locations, a lack of flexibility concerning changing requirements, and, in the event of a failure, long downtimes of the system [103,105].

*UAV (A4).* UAVs are defined as cargo aircrafts that operate autonomously without pilots. The main benefits of their implementation are reflected in 3D navigation for MH activities [106–108]. Although UAVs are effectively used for control and logistics activities throughout the SC, they are particularly suitable to increase the effectiveness of logistics processes, especially the transportation of goods within the warehouse. UAVs usually function flawlessly in narrow and high warehouse corridors [106,109]. The use of UAVs is notably effective for the implementation of intralogistics activities within production systems. This alternative contributes to saving time and space in the warehouse, which contributes to its application in inventory management [108–111]. With the application of AI technology, UAVs can be operated automatically and without the need for direct human supervision. Navigation and management are performed using CGPS systems and IIoT sensors that provide the necessary data on the positions of equipment and drones [79,83]. CWMS is necessary for the communication of all MH and other equipment and goods with the central computer. CWMS should provide information on inventory status, the planning and scheduling of an assignment to a UAV, assignment location, arrival time and the positions of external transport vehicles, the execution of an assignment and order status [85]. The necessary data on the locations of goods, employees and equipment are obtained using IIoT sensors and are stored using BD technology [80,86,88]. In addition, IIoT collects data on the readiness of goods for unloading and storage and on the time and place of positioning the external transport vehicles and UAVs, thus preventing crashes of UAVs and collisions with other equipment [79,84]. The collected data are processed and analyzed with DM technology, and thus, the sequence of serving the storage locations and the service time are determined. Moreover, DM has an essential role in UAV routing, thus achieving high security at the warehouse location where goods are picked and delivered [86,88,107,111].

However, for this alternative to be effective, clearly defined laws and regulations are needed, as well as a developed awareness of acceptability among employees [87].

The possibility of continuously and precisely performing intralogistics activities, reduced operating costs, minimized maintenance costs and increased safety in logistics centers, are the main advantages of UAVs [109–112]. This alternative is environmentally and socially sustainable because it is electrically powered and does not require drivers [107,108,111]. Some limitations relate to technical characteristics, such as a low payload, limited range and frequent battery charging requirements. In addition, significant initial investments call into question economic sustainability. Therefore, they are often used in combination with other autonomous alternatives [107–110,112].

### 4.2. Evaluation Criteria for Smart MH Solutions

The optimal alternative solution is selected using the MCDM method, which necessitates the formulation and selection of relevant criteria for evaluating proposed alternatives. To select the optimal MH equipment, criteria are grouped into three groups: technical–technological, economic and normative. The criteria are defined based on numerous previous studies in the field of MH equipment selection. Below is a description of the defined criteria.

Technical–technological criteria:

*Efficiency (C1)* represents the total time that is required for the implementation of logistics activities per order, from the moment of receiving the order to the moment of delivery the goods. The alternative that requires less time to complete the task is preferred. [113–115].

*Technological development (C2)* suggests the degree of technological development, which implies the level of its application in practice. The alternative that is more often used in practical applications is favored [116].

*Utilization complexity (C3)* includes the time taken to implement the technology and the ability to integrate the technology with other or existing systems. It assumes the ability to connect with existing information systems, such as CWMS, IIoT, sensors, etc. A technology that requires less time for implementation is favored over others [113].

*Impact on MH system resilience (C4)* represents the velocity of technology's response to unpredictable circumstances (cancellation, increase in volume requirements and the growth of the warehouse system). Accordingly, technology that has the quickest response time is more favorable [112].

*Smart handling (C5)* represents the consolidation of MH activities to reduce or eliminate the engagement of employees. Contrary to traditional MH technologies, which imply the participation of employees in practically all MH activities, progressive technologies based on I 4.0 partly or completely eliminate such requirements. Therefore, technology that can independently perform several operations is more favorable [117].

*Energy consumption (C6)* is the total amount of energy that is required for MH equipment to finish a task. Therefore, technologies with lower energy consumption are more favorable [112,117].

Economic criteria:

*Procurement costs (C7)* represent the initial investments for the implementation of the alternative, including the expenditures for adapting the technology to the existing ones. According to this criterion, technology that requires less investment is more favorable [112,114].

*Labor cost savings (C8)* represent lowering the costs for employee wages. An alternative that can perform multiple operations independently requires fewer employees, thus reducing costs. In addition, it also includes the difference in the total costs of operating the new technology system in relation to the operation of the existing system [113].

*Maintenance costs (C9)* include the costs of servicing, infrastructure maintenance, etc. Lower costs indicate that technology is more favorable regarding this criterion [112,118].

*Return on investment (C10)* indicates the time that is required for a return on an investment. A shorter period suggests that the alternative is more favorable regarding this criterion [113,119].

Normative criteria (standardization and regulation):

*Employee safety (C11)* indicates the adopted standards on occupational safety and health (IMS, ISO 45001), and the degree of their respect. Standard adoption can reduce the number of injuries and their severity. Technology that has a higher degree of automation is safer [118].

*Warehouse safety impact (C12)* refers to employee injuries and damage to goods, storage equipment and other tools, devices and MH elements. Sophisticated I 4.0 technologies are mostly automated, so an alternative that communicates more with the environment and is resistant to human error is more favorable [118].

*Standardization (C13)* indicates the adaption of existing local norms to global regulations to improve the operational environment. The alternative that has the most elevated degree of compliance with global standards is more favorable regarding this criterion [112,120].

*Employee perception (C14)* implies the ability to adopt a new alternative, which assumes employees' awareness of the acceptance of new technologies. Alternatives that are more frequently used are mainly based on modifying current (traditional) ones and are more accepted by employees [120].

### 4.3. Results of the Model's Application

The problem is hierarchically structured according to the FAHP method, as explained in Section 3. Alternative solutions that need to be evaluated are defined in Section 4.1, and the criteria according to which the evaluation is performed are described in Section 4.2. In the following, the results that are obtained by solving the problem of evaluation and the ranking of smart solutions are presented.

For the pair-wise comparison within the FAHP method, the linguistic evaluations given in Table 2 are used. The result is the formation of a matrix (1). A triangular fuzzy number (2) is defined for each criterion. The associated linguistic evaluations are presented in the following tables. Pair-wise comparisons of all criteria and groups of criteria are presented in Tables 2–5. The evaluation was conducted by experts in the field.

**Table 2.** Pair-wise comparison of technical–technological criteria.

|      | C1  | C2  | C3  | C4  | C5  | C6  |
|------|-----|-----|-----|-----|-----|-----|
| C1   | /   | RP  | MP  | QP  | QP  | SP  |
| C2   |     | /   | BP  | MP  | MP  | PP  |
| C3   |     |     | /   | BP  | RP  | QP  |
| C4   |     |     |     | /   | RP  | MP  |
| C5   |     |     |     |     | /   | QP  |
| C6   |     |     |     |     |     | /   |

**Table 3.** Pair-wise comparison of economic criteria.

|      | C7  | C8  | C9  | C10 |
|------|-----|-----|-----|-----|
| C7   | /   | RP  | QP  | PP  |
| C8   |     | /   | MP  | PP  |
| C9   |     |     | /   | RP  |
| C10  |     |     |     | /   |

**Table 4.** Pair-wise comparison of normative criteria.

| | C11 | C12 | C13 | C14 |
|---|---|---|---|---|
| **C11** | / | BP | QP | SP |
| **C12** | | / | MP | PP |
| **C13** | | | / | RP |
| **C14** | | | | / |

**Table 5.** Pair-wise comparison of criteria.

| | K1 | K2 | K3 |
|---|---|---|---|
| **K1** | / | BP | QP |
| **K2** | | / | MP |
| **K3** | | | / |

Using the FAHP method with the input data given in Tables 2–5, the criteria weights are obtained by applying Equation (7). Crisp values of the criteria weights are given in Table 6.

**Table 6.** Criteria weight via the fuzzy AHP method.

| Ci | C1 | C2 | C3 | C4 | C5 | C6 | C7 | C8 | C9 | C10 | C11 | C12 | C13 | C14 |
|---|---|---|---|---|---|---|---|---|---|---|---|---|---|---|
| Wj | 0.4 | 0.133 | 0.033 | 0.027 | 0.033 | 0.006 | 0.22 | 0.073 | 0.018 | 0.004 | 0.031 | 0.015 | 0.05 | 0.002 |

The linguistic evaluations of the alternatives according to the criteria are given in Table 7. The values used in the framework of the FCOBRA method are defined on the same fuzzy sets as those in the FAHP method (Table 1). Accordingly, the matrix of the evaluation of alternatives by criteria is formed by applying Equation (9).

**Table 7.** Evaluation of alternatives by criteria.

| | A1 | A2 | A3 | A4 |
|---|---|---|---|---|
| **C1** | SB | RB | VB | BB |
| **C2** | AB | QB | QB | RB |
| **C3** | AB | QB | BB | RB |
| **C4** | VB | MB | EG | RB |
| **C5** | SB | MB | SB | BB |
| **C6** | MB | SB | BB | QB |
| **C7** | PB | VB | EG | RB |
| **C8** | SB | MB | PB | RB |
| **C9** | VB | QB | EG | MB |
| **C10** | AB | PB | BB | MP |
| **C11** | MB | QB | SB | PB |
| **C12** | QB | QB | PB | RB |
| **C13** | AB | SB | QB | EG |
| **C14** | AB | RB | QB | EG |

The criteria weights (Table 6) that are obtained via the FAHP method from applying Equation (15), in combination with the evaluations of the alternatives (Table 7), represent the input data for the FCOBRA method. The final ranking of the alternatives that are obtained by applying the FCOBRA method (31) is presented in Table 8. The obtained results show that the best-ranked alternative is A1, and the worst-ranked alternative is A4. In second place is A3, and in third place is A2.

**Table 8.** Optimal MH equipment ranked by fuzzy COBRA method.

|          | Norm   | Rank |
|----------|--------|------|
| **dC (A1)** | −0.070 | 0.00 | 1 |
| **dC (A2)** | 0.024  | 0.56 | 3 |
| **dC (A3)** | −0.050 | 0.12 | 2 |
| **dC (A4)** | 0.097  | 1.00 | 4 |

*4.4. Sensitivity Analysis*

To determine whether the obtained solution is resistant to changes in the model setup, a sensitivity analysis is performed. The results that are obtained in Section 4.3 are adopted as the basic sensitivity analysis scenario (Sc. 0). Within the sensitivity analysis, five different scenarios are defined. In each of them, some change is made regarding the weight or consideration of some criterion. In Sc. 1, all criteria weights are equalized. In Sc. 2, 3 and 4, criteria C1, C7 and C11 are excluded from the model, respectively. The criteria that are excluded from the model are the ones with the highest weights in their criteria group. In Sc. 5, all three mentioned criteria are excluded.

Table 9 demonstrates the results of the sensitivity analysis. In all scenarios, A1 is the best-ranked, and A4 is the worst-ranked alternative. In Sc. 1 and Sc. 5, the alternative A3 is ranked second. In other Sc. (2, 3 and 4), the A4 alternative is second.

**Table 9.** Sensitivity analysis results.

|          | Sc. 0  | Sc. 1  | Sc. 2  | Sc. 3  | Sc. 4  | Sc. 5  |
|----------|--------|--------|--------|--------|--------|--------|
| **dC (A1)** | −0.070 | −0.218 | −0.064 | −0.082 | −0.122 | −0.093 |
| **dC (A2)** | 0.024  | −0.012 | −0.009 | 0.016  | −0.005 | −0.009 |
| **dC (A3)** | −0.050 | 0.069  | −0.037 | −0.076 | −0.017 | 0.007  |
| **dC (A4)** | 0.097  | 0.183  | 0.036  | 0.056  | 0.078  | 0.050  |

The results of the sensitivity analysis (Figure 1) demonstrate that the defined methodology is stable enough, i.e., that the obtained solution is credible. This is supported by the fact that, in all defined scenarios, A1 is the best-ranked alternative.

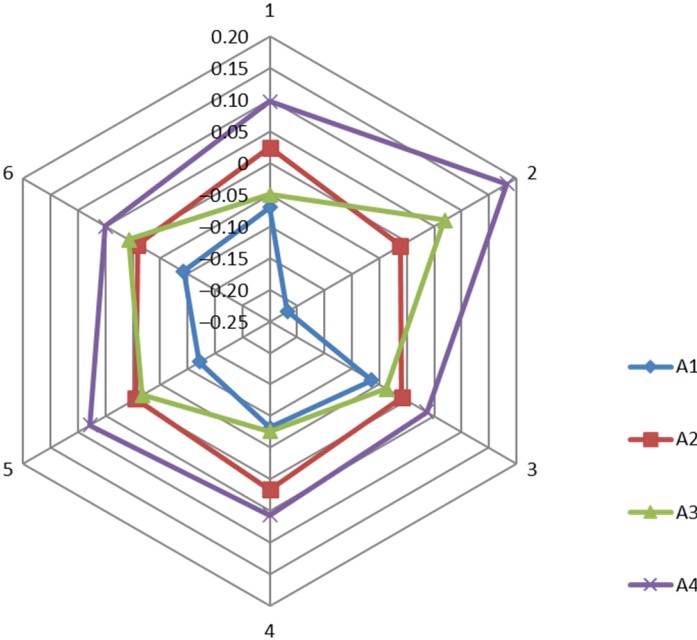

**Figure 1.** Sensitivity analysis results.

*4.5. Validation of Results*

To validate the obtained results, the problem with the same input values is also solved by applying several others; in the literature, the most commonly used distance-based MCDM methods are TOPSIS, VIKOR, CODAS, EDAS, MOORA, Weighted Aggregated Sum Product Assessment (WASPAS), and Measurement Alternatives And Ranking According To Compromise Solution (MARCOS). The obtained rankings of the alternatives are shown in Table 10. As a measure of the similarity of the ranks obtained using other methods with the rank obtained using the COBRA method, the Spearman correlation coefficient (SCC) is calculated. The mean SCC value of 0.92 indicates that there is a very high degree of agreement between the results obtained by the COBRA method and the results obtained using other methods. In addition, all methods imply that the same alternative is the best. Based on the above, it can be concluded that the obtained solution is valid, i.e., that the selected alternative is the most favorable concerning the observed criteria.

**Table 10.** Validation of results.

| Rank | COBRA | TOPSIS | VIKOR | CODAS | EDAS | MOORA | WASPAS | MARCOS |
|------|-------|--------|-------|-------|------|-------|--------|--------|
| A1 | 1 | 1 | 1 | 1 | 1 | 1 | 1 | 1 |
| A2 | 3 | 3 | 2 | 3 | 2 | 3 | 2 | 3 |
| A3 | 2 | 2 | 3 | 2 | 3 | 2 | 3 | 2 |
| A4 | 4 | 4 | 4 | 4 | 4 | 4 | 4 | 4 |

## 5. Discussion

In MH activities, there are always numerous challenges that significantly affect the preparation of goods for delivery and therefore the efficiency of customer service. Given the prevalence of MH activities in logistics centers, companies are forced to employ modern technologies to integrate logistics activities with stochastic customer requirements. Taking into account the mentioned criteria as well as the data from the case studies, alternative solutions are ranked in the paper to increase the efficiency of the MH process in the logistics center, which leads to the satisfaction of the demands of users in the city. According to the ranking of the alternatives, implementing A1 is the most appropriate replacement for the fleet of electric forklifts in the observed logistics company. The engagement of A1 can bring numerous benefits, such as high positioning accuracy, which can reduce damage to goods and equipment, and the complete automation of logistics activities, which leads to reductions in employee errors. Minor modifications of the current system are needed for implementing A1 compared to the other alternatives, because the same type of MH equipment as that in the observed system is involved. Other benefits include a high degree of employee acceptance and increased safety of the entire system. All the advantages brought by implementing A1 can lead to more efficient order fulfillment compared to the current system. Faster execution of the order, while reducing errors and damage to goods, can lead to the timely and accurate delivery of goods in the city. Optimizing the MH process using A1 can reduce negative effects on the delivery of goods in the city, such as defective goods and delivery delays, and can reduce the number of total drives and, therefore, GHG emissions. The sensitivity analysis confirms that A1 is the optimal solution.

According to the available data and defined criteria, alternative A1 is suggested as the most favorable one in the observed case study. The implementation of the contemporary fleets that are implied in this alternative is the most appropriate replacement for the fleet of electric forklifts in the observed logistics company. A1 engagement provides numerous benefits in the sense of minimizing errors in MH activities, reducing the duration of order fulfillment and incurring insignificant expenses for implementation in the current layout. The second-ranked alternative is A3. Although it increases the utilization of space, this alternative requires high investments and lacks sufficient flexibility when changing customer requirements. Despite the high degree of precision and a brief order fulfillment time, A3 requires high automation of the entire logistics center, which is an additional

limitation in its implementation. A2 is third because cargo cannot be actively picked-up and disposed of. As a consequence, additional investments in MH automation are required, which affects the reduction in transport capacity and movement speed as well as the extension of battery charging time. Despite their growing importance in logistics and control activities, A4 has a limited payload and range, which puts it in the fourth position. Difficult acceptance by employees, as well as large investments and low battery capacity, further contribute to the rank of A4. Considering its growing role in intralogistics processes, A4 should be analyzed in combination with other smart alternatives.

The field of MH has always been exposed to numerous challenges that particularly affect the efficiency of customer service in the city. As most of these activities are performed within logistics centers, logistics companies are forced to engage in modern technologies to integrate logistics activities with stochastic customer requirements. Respecting the pre-defined criteria as well as the data from a real case study, this paper ranks the alternative solutions to increase the efficiency of MH in logistics centers. This paper presents the application of the new MCDM model, which is based on a combination of the FAHP and FCOBRA methods. FAHP is used to determine the relative weights of the criteria, and FCOBRA is used to obtain the ranking of the alternatives. The motive for developing a new novel MCDM is the type of solved problem, which requires the observation of several groups of different and conflicting criteria. Based on that, the need for a multi-criteria analysis arises. The main motivation for using the FAHP approach is that the structure of the method corresponds to the type of problem being solved. The problem can be structured so that the goal—the selection of smart MH solutions—is at the top, with the criteria at the next level and finally with the alternative at the bottom. The main disadvantage of the FAHP method is its complexity and the need for pair-wise comparisons at all levels of the problem structure. Therefore, the COBRA method is used in the second part of the model, which significantly reduces the number of comparisons. The FCOBRA method takes into account different metrics for calculating the distance from the relevant solutions, which could also be a disadvantage, i.e., the use of the method is somewhat complex.

The implications of this paper are multiple. From the aspect of methodology, the theoretical implication is that, for the first time, a combination of FAHP and FCOBRA is performed in research, thus enriching the existing literature. The practical implication of the defined methodology is that it can be used as a DSS for real problems of a similar nature. Regarding the considered problem, the theoretical implication is the definition of smart MH solutions that combine modern MH equipment with I 4.0 technologies and thereby overcome certain shortcomings of traditional MH solutions. The practical implication of the solutions is reflected in the fact that they can be applied to similar systems and logistics centers of different categories to improve the efficiency and performance of both the centers themselves and the city logistics of the analyzed area as a whole. Potential limitations may be related to the number and type of alternatives and the set of selected criteria. The limitation of the used MCDM model is reflected in its complexity and robustness, which implies the engagement of significant resources (time, human, financial, etc.).

## 6. Concluding Remarks

Due to increased urbanization, the volume of e-commerce and the emphasis on online ordering, the delivery of goods has become the most vulnerable part of CL. Therefore, logistics companies face numerous challenges in logistics centers as entities that coordinate intralogistics activities and stochastic customer demands in the city. It is recognized that the activities of MH significantly affect the efficiency of serving users as well as all pillars of CL sustainability, which require the engagement of modern smart technology based on I 4.0.

One of the approaches to overcoming the main challenges in the preparation of goods for delivery in logistics centers is the optimization of MH activities through engagement with autonomous smart solutions. Namely, the logistics company, whose activity is the processing and delivery of consumer goods in the city, uses electric forklifts for performing MH activities.

Bearing in mind the numerous analyzed limitations, the overcoming of which would increase the efficiency of customer service, the main goal of this paper is the selection of smart solutions for MH based on combinations of modern MH equipment and I 4.0 technology. In order to overcome some key challenges, four alternatives are discussed in this research: SAGV forklifts, AMRs, GTP and UAVs, which are supported by CWMS, CGPS, BD, DM, IIoT, AI and AR as the most significant I 4.0 technologies for MH activities [81–89,95–108]. Each alternative is analyzed in terms of its advantages and disadvantages from the perspective of MH in logistics center and are evaluated based on fourteen criteria. The observed criteria are divided into three groups: technical–technological, economic and normative. For the evaluation of these alternatives by the defined criteria, an innovative combination of the FAHP and FCOBRA methods is developed in this paper. FAHP is used to determine the criteria weights, and then the alternatives are ranked using the FCOBRA method. According to the analyzed criteria, A1 represents the most sustainable alternative for the observed case study. This solution requires minor modifications to the existing MH system in the logistics center of the analyzed company. It also implies a high degree of automation for the logistics process, increased safety at work (employees, goods, equipment, etc.), lower costs (procurement, work, maintenance, etc.), improved warehouse space utilization of roughly 35% when compared to conventional electric forklifts, 2–3 times and 10 times more payload compared to A2 and A4, respectively, and other advantages that favor it compared to other alternatives. Sensitivity analysis and validation of the results indicate that the obtained solution is relevant and valid.

By analyzing the papers in the field of CL, some of the numerous challenges in the delivery of goods in the city that occur in logistics centers and that significantly affect the (in)efficiency of customer services are identified. Respecting the mentioned circumstances, in this research, the FAHP and FCOBRA methods are combined to improve MH activities in logistics centers, which can significantly affect the increase in the resilience of the entire CL. Given that this research was performed based on relevant data from the work of one of the largest German logistics companies for customer service in the city, this paper provides adequate guidelines for other logistics companies that face similar challenges to improve the efficiency of customer service.

Some of the key limitations of this research are related to the integration of at least two smart alternatives for smart MH activities in the logistics center, as well as the definition of other criteria that support the importance of the decisions that are made. Practical gaps in the implementation of the obtained solutions could be related to the limitation of resources in the implementation of modern I 4.0 technologies. As modern MH alternatives require the informational readiness of all SC participants, one of the key challenges is the development and implementation of appropriate applications that enable more efficient implementation of customer requests. The complexity and robustness of the applied MCDM model represent the limitations of its application, which requires further investigation.

**Author Contributions:** Conceptualization, S.T., M.K., S.D.-M. and M.B.; Formal analysis, S.T., M.K., S.D.-M. and M.B.; Methodology, M.K. and M.B.; Writing—original draft preparation, S.T., M.K., S.D.-M. and M.B. All authors have read and agreed to the published version of the manuscript.

**Funding:** This research received no external funding.

**Institutional Review Board Statement:** Not applicable.

**Data Availability Statement:** The used data are owned by the company. They may be made available with the company's permission.

**Conflicts of Interest:** The authors declare no conflict of interest.

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
