# Peer review of "Smart Material Handling Solutions for City Logistics Systems"

_sustainability, doi:10.3390/su15086693_

Round 1
Reviewer 1 Report
The paper faced the problem of ranking and selecting smart material handling solutions for distribution systems by combining FAHP and FCOBRA multi-criteria decision making methods.
I would ask authors to address the specific comments, as follows:
1. Please consider to change the title since the paper aims to select smart material handling solution for distribution system.
2. Please add a couple of more sentences that justify the application of the FAHP and FCOBRA methods with regards to other multi-criteria decision methods for material handling equipment selection.
Minor comments
Time windows would be more appropriate term than “time frames” or “windows frame”
Line 215: should is missing in “MCDM methods be used”
Author Response
Dear reviewer,
We thank you for the constructive and useful comments which helped us improve the quality of our paper. We have acknowledged all comments and corrected the paper according to them. We have provided point-to-point responses to your comments and highlighted the changes in the manuscript (we used Track changes mode).
Comment 1: Please consider to change the title since the paper aims to select smart material handling solution for distribution system.
Response: We have changed the title of the paper. However, we did not relate it to the distribution system but to the city logistics system. We find it more appropriate and more in line with the changes we made in the paper (required by the other reviewers).
Comment 2: Please add a couple of more sentences that justify the application of the FAHP and FCOBRA methods with regards to other multi-criteria decision methods for material handling equipment selection.
Response: We have additionally justified the application of the FAHP and FCOBRA methods (in the last two paragraphs of the sub-section 2.4)
Comment 3: Time windows would be more appropriate term than “time frames” or “windows frame”
Response: Thank you for noticing this. We have changed both terms into "time windows" throughout the paper.
Comment 4: Line 215: should is missing in “MCDM methods be used”
Response: We have done extensive English proofreading and eradicated this and other similar errors.
Reviewer 2 Report
How to deploy the distribution centers of delivery goods is an important issue and a highly vulnerable part of the supply chain as they integrate manufacturing activities and stochastic customer demands. To this end, the authors utlized 4 smart solutions and 14 evaluation criteria to present a new hybrid Multi-Criteria Decision-Making model, which combines the Fuzzy AHP method and the Fuzzy COmprehensive distance-Based RAnking method. As a further step, they devised an autonomous forklift that could greatly automate logistics activities and reduce the rate of delivery errors.
In my opinion, the idea could be interesting, and I suggest the potential acceptance after some necessary revision as follows
1) The motivation needs to be further refined, and the current edition is a little vague;
2) The results should be compared with other typical methods;
3) 14 evaluation criteria are selected, and it could be improved since too many criteria are difficult to be resolved or anlyzed in one model;
4) The words and expressions can be further polished. As an example, in Section 2, "...realizing the user's request" -->"...meeting the user's request" could be better.
Author Response
Dear reviewer,
We thank you for the constructive and useful comments which helped us improve the quality of our paper. We have acknowledged all comments and corrected the paper according to them. We have provided point-to-point responses to your comments and highlighted the changes in the manuscript (we used Track changes mode).
Comment 1: The motivation needs to be further refined, and the current edition is a little vague.
Response: We have refined the motivation by correcting the Abstract, Introduction, literature review and discussion.
Comment 2: The results should be compared with other typical methods.
Response: The results have been compared to seven other MCDM methods. We have added an entirely new sub-section dealing with the validation of results (sub-section 4.5).
Comment 3: 14 evaluation criteria are selected, and it could be improved since too many criteria are difficult to be resolved or anlyzed in one model;
Response: We agree that 14 criteria are difficult to address in one model. We have highlighted that in the limitations of the study. However, there are multiple examples in the literature where problems with even larger number of criteria were successfully solved using the AHP method. Some of the most recent examples are:
- Awad, J., & Jung, C. (2022). Extracting the planning elements for sustainable urban regeneration in Dubai with AHP (analytic hierarchy process). Sustainable Cities and Society, 76, 103496.
- Hossain, M. K., & Thakur, V. (2021). Benchmarking health-care supply chain by implementing Industry 4.0: a fuzzy-AHP-DEMATEL approach. Benchmarking: An International Journal, 28(2), 556-581.
- Khan, A. U., & Ali, Y. (2020). Analytical hierarchy process (AHP) and analytic network process methods and their applications: a twenty year review from 2000-2019: AHP & ANP techniques and their applications: Twenty years review from 2000 to 2019. International Journal of the Analytic Hierarchy Process, 12(3).
In addition, we believe that we have managed to do the same and that the results we obtained are relevant, therefore we did not make any changes regarding the number of criteria.
Comment 4: The words and expressions can be further polished. As an example, in Section 2, "...realizing the user's request" -->"...meeting the user's request" could be better.
Response: We have done extensive English proofreading and eradicated this and other similar errors.
Reviewer 3 Report
Thanks for submitting this paper for being considered in Sustainability. The manuscript (sustainability-2312092) is empirical study aiming to rank and select smart material handling solutions for distribution systems. The topic addressed is worth of investigation, and the theoretical/empirical approaches considered result interesting and may contribute in a substantial manner to the scientific state of city logistics.
My comments are listed below:
The paper is generally well-written and analyses appear to be appropriate. Although the authors conducted exhaustive literature review, it is not entirely clear to this reviewer how the current study makes a significant contribution above other papers. The authors should be more clear about the relevance of their research for the problem of city logistics. In my oppinion, the introduction may be further clarified and strengthened if the authors describe in a more structured way in what respects their research replicates earlier research. This is one issue that the authors should describe more clearly in the introduction (and in the discussion). I believe that the introduction and discussion should be written more to the point.
Author Response
Dear reviewer,
We thank you for the constructive and useful comments which helped us improve the quality of our paper. We have acknowledged all comments and corrected the paper according to them. We have provided point-to-point responses to your comments and highlighted the changes in the manuscript (we used Track changes mode).
Comment 1: The paper is generally well-written and analyses appear to be appropriate.
Response: We thank the reviewer for this affirmative comment.
Comment 2: Although the authors conducted exhaustive literature review, it is not entirely clear to this reviewer how the current study makes a significant contribution above other papers. The authors should be more clear about the relevance of their research for the problem of city logistics.
Response: We have entirely rewritten sub-section 2.1, dealing with the city logistics. We have established a clear connection between the problem and the city logistics.
Comment 3: In my oppinion, the introduction may be further clarified and strengthened if the authors describe in a more structured way in what respects their research replicates earlier research. This is one issue that the authors should describe more clearly in the introduction (and in the discussion). I believe that the introduction and discussion should be written more to the point.
Response: We have restructured the Introduction and rewritten most of the first part of this section. In addition, we have made specific changes to the Abstract according to the changes made in the Introduction and literature review. We have also strengthened the Discussion. We highlighted the significance of the obtained results and theoretical and managerial (practical) implications.